# Application of Instrumented Indentation Procedure in Assessing the Low-Cycle Fatigue Properties of Selected Heat-Treated Steels

**DOI:** 10.3390/ma17102375

**Published:** 2024-05-15

**Authors:** Bogusław Hościło, Krzysztof L. Molski

**Affiliations:** 1Faculty of Mechanical Engineering, Bialystok University of Technology, Wiejska 45C, 15-351 Bialystok, Poland; 2SaMASZ Sp. z o.o., ul. Trawiasta 1, 16-060 Zabludow, Poland; krzysztof.molski@samasz.pl

**Keywords:** fatigue of metals, strain-controlled fatigue characteristics, depth sensing hardness, fatigue life assessment

## Abstract

The paper presents an analysis of the low-cycle fatigue (LCF) properties of C45, X20Cr13, and 34CrNiMo6 steels subjected to various heat treatment processes. Strain-controlled LCF tests were carried out with a total cyclic strain amplitude equal to 0.5, 1 and 1.5%. Fatigue life, cyclic stress-strain behavior and hardness were analyzed. Qualitative and quantitative relationships between material LCF properties resulting from the heat treatment processes, were related to the indentation force *P**, which was derived experimentally by applying an instrumented indentation procedure with the use of the Vickers indenter. The proposed parameter *P** and its changes Δ*P** seem to be promising for the identification of the structural stress parameter *σ** that is necessary for deriving values of the fatigue strength coefficients *σ_f_’* corresponding to different tempering temperatures. The common feature of all steels analyzed in this paper is that the elastic parts of the strain-life characteristics remain parallel after being subjected to different tempering temperatures.

## 1. Introduction

Machine elements and structures are usually subjected to variable loads that may cause fatigue damage. The design of such mechanical parts consists in the preliminary assessment of fatigue life and requires the experimental representative fatigue data of particular materials [1,2]. Such material characteristics are valid for a given or supposed material state usually considered in the design [3,4]. In many cases, modern technology requires versatility in material features, which may be formed by different processing methods. In this way, many mechanical properties, such as yield and ultimate stress, hardness, elongation, fracture toughness, high-cycle fatigue characteristics etc., may be obtained by a properly applied technological process [5]. Many types of steels are suitable for heat treatment, which is generally understood as heating to the strictly defined temperature, quenching and tempering [6,7,8]. It is well known that such processes may significantly change all mechanical properties including fatigue strength [9,10]. However, there is a lack of information about the influence of the heat treatment process on qualitative and quantitative changes in the low-cycle fatigue (LCF) characteristics of steels, especially when the material is subjected to different tempering temperatures. From the point of view of the designer, the knowledge of how low-cycle fatigue characteristics change would be valuable. An example of an experimental program of low-cycle fatigue for various specimen geometries and loading conditions is described in [11].

Indentation methods have been used for many years to determine the mechanical properties, including fatigue behavior, of various materials. Attempts to find relationships between the hardness of materials and their mechanical properties are known in the literature, as hardness is a convenient and relatively easy parameter to measure. In paper [12], the authors searched for a relationship between material surface hardness and fatigue strength. The authors of paper [13] proved that changes in surface hardness obtained by using a pack-carburizing process showed that the changed fatigue behavior depended on the heat treatment and that it can improve fatigue life. In paper [14], the authors proposed a linear relationship between the fatigue limit and the Brinell Hardness for selected steels and aluminum alloys. However, in the available literature, there is a lack of works dealing with direct relationships between the quantities obtained in the instrumental indentation process and material properties in terms of LCF. Knowledge of such relationships would allow for an easier estimation of fatigue life based on relatively simple experimental tests.

The present paper deals with the influence of the heat treatment process on the strain-controlled low-cycle fatigue behavior of steels with different mechanical properties and chemical compositions. The assumption was made that the tempering temperature of the heat treatment process produces qualitative and quantitative changes in LCF material characteristics [15,16]. However, some more general conclusions arising from the fatigue behavior of steels in such conditions may be drawn after comparing the experimental results of more than one material. For this reason, three different steels, namely: C45, X20Cr13, and 34CrNiMo6 were chosen. On the other hand, it is convenient to have a tool for identifying material features corresponding to LCF properties after a heat treatment process. One approach, presented in [17], applied to X20Cr13 steel, seems to be very promising. It is based on Vickers probe together with the instrumented indentation procedure for a constant indentation depth *h** = 4 μm. The method has appeared to be very effective in determining the magnitude of the equi-biaxial stresses at the surface layer of the X20Cr13 material and may be especially useful in identifying residual stresses. Constant indentation depth *h** makes it possible to measure the strength of material, represented physically by external force *P**, and maintaining similar penetration conditions while applying the probe to a given depth. The same measurement procedure, based on the *P** force, has been applied to identify the strength of materials after various heat treatment processes.

The main objective of the present work was to experimentally determine the influence of the heat treatment process on LCF characteristics for C45, X20Cr13, and 34CrNiMo6 steels and to analyze the possible applicability of indentation force *P** as a potential discriminant parameter of material features.

## 2. Materials, Specimens and Heat Treatment

Three steels, C45, X20Cr13, and 34CrNiMo6, suitable for quenching and tempering, were chosen. These steels of different mechanical properties and chemical compositions are commonly used in mechanical engineering. The mechanical properties of the selected steels are given in Table 1.

Fatigue properties and failure mechanisms under different loading conditions are presented in [19,20,21,22,23].

Each material was delivered in the form of 16 mm-diameter cylindrical cold drawn bars. The chemical composition of all materials, determined using a Thermo ARL Quantris optical emission spectrometer (Thermo Fisher Scientific Inc., Waltham, MA, USA) and 10 mm-thick samples, are presented in Table 2.

Geometrical details of the specimen are shown in Figure 1. Conical ends provided co-linear fixing, high rigidity and prevented the specimens from sliding in the jaws of the hydraulic machine when testing the very hard material. All specimens were placed in the testing machine using specially designed and fabricated mounting devices. A grinding allowance, left in the central part of all specimens, made it possible to prepare the surface properly for LCF testing.

Before performing the heat treatment processes on particular specimens, preliminary tests of all materials were carried out in order to relate material hardness to the tempering temperatures. Three different hardness levels were chosen. The procedure was carried out according to the information concerning quenching temperatures and liquids provided by the steel producers. The heat treatment parameters and hardness measurements obtained from the preliminary tests are shown in Table 3.

The process was performed in a furnace with a protective atmosphere. Hardness measurements were made using Rockwell and Vickers’ methods on the end faces of the specimens.

Figure 2a–c shows the microstructures of C45 steel samples quenched from 850 °C and (a) tempered at 620 °C, (b) at 520 °C and (c) at 620 °C. Details of the heat treatment are shown in Table 3. The annealing time before quenching was 45 min and the tempering time was 20 min. The microstructure of a sample after tempering consists mainly of martensite, pearlite, some ferrite and a mixture of ferrite and cementite with a dispersion that decreases with increasing temperatures. This results in a decrease in hardness and an increase in ductility of the material [24,25].

Figure 2d–f shows the microstructure of the martensitic X20Cr13 stainless steel consists of ferrite and spherical carbides in the annealed state. After heat treatment (austenitising at 1000 °C for 45 min and tempering at 700 °C, 620 °C and 550 °C for 20 min), low-carbon tempered martensite is the main phase in the microstructure [26,27].

34CrNiMo6 steel met the performance requirements after appropriate heat treatment consisting of quenching and tempering at high temperatures [28]. After oil quenching of the samples (heated at 850 °C for 45 min) and tempering at 700, 620 and 500 °C for 20 min by air cooling, the microstructure of the samples was mainly homogeneous tempered martensite (Figure 2g–i).

## 3. LCF Tests and Results

### 3.1. Experimental Procedure

All strain-controlled LCF tests were carried out at room temperature with a strain ratio of *R_ε_* = *ε*min/*ε*max = −1 and with a frequency of 0.2 Hz, using an Instron 8502 hydraulic machine (Instron, High Wycombe, UK) provided with an extensometer with a 12.5 mm gauge length. Three different values of total strain amplitude *εa* equal to 0.5%, 1.0% and 1.5% were chosen. Such values were considered as the most representative for analyzing LCF properties of the materials subjected to heat treatment due to the fact that both elastic and plastic strain amplitudes are of the same order. All tests were terminated when the force dropped to 50% of the force corresponding to 0.5 *Nf* [29].

### 3.2. Cyclic Stress–Strain Behavior

Hysteresis loops for the half-life cycle for all chosen strain amplitudes and materials subjected to heat treatment procedures were identified. Some examples of the experimental results for the strain amplitude *ε_a_* = 1.5% are shown in Figure 3. Additionally, the particular data corresponding to all LCF tests are presented in Table 4, Table 5 and Table 6. All remaining experimental data, including first, half-life, and final hysteresis loops, for total strain amplitudes *ε_a_* equal to 0.005, 0.010, and 0.015, are given in Appendix A. The maximum cyclic stress *σ*_max_ history, representing the material degradation process during cyclic loading, is presented in Figure A10 for *ε_a_* = 1.5%. The fracture surfaces of the fatigue-tested specimens, depending on their hardness and strain amplitudes, are shown in Table A1. It can be observed that the brittle fracture corresponds to lower strain amplitudes and higher hardness of the specimens. Because the tests were conducted in the elastic–plastic strain range, the fractures are dominated by plastic deformation.

### 3.3. Cyclic Stress–Strain Curves

Graphical representations of the cyclic stress–strain characteristics obtained from the experimental results of stabilized hysteresis loops are shown in Figure 4.

These curves were derived using the Ramberg–Osgood formula [30] given by Equation (1) by fitting experimentally obtained values of stress and strain ranges to the formula as follows:(1)$$\frac{\Delta \varepsilon }{2}=\frac{\Delta \varepsilon_{e}}{2}+\frac{\Delta \varepsilon_{p}}{2}=\frac{\Delta \sigma }{2E}+{\left(\frac{\Delta \sigma }{2K^{\prime }}\right)}^{1/n^{\prime }}$$

Particular values of *K*′ and *n*′, corresponding to the Ramberg–Osgood equation for different hardness ranges of particular materials are presented in Table 7.

### 3.4. Strain–Life Characteristics

Other sets of experimental results, obtained from LCF tests, are represented by strain–life characteristics based on stabilized loops for 0.5 *Nf* and depicted for particular materials in Figure 5, Figure 6 and Figure 7. For convenience, corresponding pairs of elastic and plastic strain amplitudes are shown separately. In such a way, the mutual proportions between elastic and plastic parts of strain amplitudes are clearly seen, and the Manson–Coffin–Basquin Formula (2) may be applied as the mathematical representation of experimental results [31] as follows:(2)$$\frac{\Delta \varepsilon }{2}=\frac{\Delta \varepsilon_{e}}{2}+\frac{\Delta \varepsilon_{p}}{2}=\frac{\sigma_{f}^{\prime }}{E}{\left(2N_{f}\right)}^{\;b}+\varepsilon_{f}^{\prime }{\left(2N_{f}\right)}^{\;c}$$
where *E*, *σ_f_*′, *ε_f_*′, *b*, *c* are material constants.

The values of the Manson–Coffin–Basquin parameters for all steels subjected to heat treatment are presented in Table 8.

Particular values of *b* have been found as follows: –0.0706 ± 0.0111 for C45, –0.0818 ± 0.0043 for X20Cr13 and –0.0976 ± 0.0019 for 34CrNiMo6 steel. Plastic parts of the strain–life characteristics change their position and rotate. Such a qualitative feature is common for all steels being analyzed.

### 3.5. Conclusions Arising from LCF Tests

The experimental results of LCF tests have shown that material characteristics, represented by the Ramberg–Osgood and Manson–Coffin–Basquin equations, depend on the parameters of the tempering process. For all steels: C45, X20Cr13, and 34CrNiMo6, qualitative relationships between tempering temperatures and the material characteristics were very similar. Lower tempering temperature applied in the process resulted in higher values of the cyclic stress–strain branches and the elastic part of the strain–life characteristics. This observation is rather obvious and might be expected. However, the most interesting feature of all steels analyzed here is that elastic parts of the strain–life characteristics remain parallel after being subjected to different tempering temperatures, which means that the fatigue strength exponent *b* does not change and the fatigue strength coefficient *σ*_f_’ should be modified. A similar phenomenon can be also observed in [32,33], as a result of plastically pre-strained and asymmetrically loaded steels.

Therefore for each material Equations (3) and (4) are proposed
*b*_1_ = *b*_2_ = *b*_3_ = *b*(3)
(4)$$\varepsilon_{ae}=\frac{\sigma_{f}^{\prime }-\sigma^{*}}{E}{\left(2N_{f}\right)}^{b}$$
where particular subscripts 1, 2, and 3 in Equation (3) correspond to different tempering temperatures.

The Equation (4) is similar to the Morrow fatigue formula, where the effect of mean stress on fatigue life is considered. In the present case the meaning of an additional term *σ**, necessary for correcting the value of Basquin’s strength coefficient, was named as structural stress.

## 4. Determination of Characteristic force *P** and Structural Stresses *σ**

### 4.1. Material and Method

A method of determining surface stresses in X20Cr13 steel through the use of the Vickers indenter is presented in [17]. The method consists in applying the probe to reach a characteristic constant penetration depth *h**. In this way, a characteristic stabilized force *P** was obtained that depended on the equi-biaxial stress state of the material surface. It was shown that the relationship between the equi-biaxial stresses and the differences in indentation force Δ*P** is linear.

The same approach has been used in the present study for analyzing material properties corresponding to the elastic part of fatigue characteristics, occurring in the Manson–Coffin–Basquin equation. The procedure used for determining the magnitude of the characteristic force *P** is based on the indentation method, considering a constant indentation depth *h** = 4 μm.

The specimens prepared for deriving the characteristic values *P** were made from the same materials subjected to heat treatment, with carefully finished surfaces, as shown in Figure 8.

Indentation tests were performed several times for each specimen in order to obtain average *P** values. Examples of the indentation histories are presented in Figure 9.

### 4.2. Experimental Results

Average values of characteristic stabilized forces *P** are summarized in Table 9.

It can be seen that the particular values of the indentation force *P** are different for each material and depend on the tempering temperature of the Q and T process. The relationships between the indentation force *P** and the fatigue strength coefficient *σ_f_′* for particular steels and different tempering temperatures are depicted in Figure 10a.

For each material, particular Δ*P***_i_* values were obtained by equation (5), representing the increase of the characteristic indentation force with respect to the lowest *P** value (corresponding to the highest tempering temperature), named as *P**_1_ as follows:(5)$$\Delta P_{i}=P_{i}-P_{1}$$

Next, using Equation (6), the particular values of the structural stresses *σ**, based on experimentally determined strength coefficients and indentation forces, were derived as follows:(6)$$\sigma_{i}=f\left(\sigma_{fi}^{\prime },\;\Delta P_{i}\right)$$

The relationship between *σ** and Δ*P** for particular steels are given in Table 10 and shown in Figure 10b.

## 5. Discussion

The most important observation resulting from the experimental data is that elastic parts of strain–life relationships appeared for the elastic strain amplitudes, being parallel for each material. This means that the fatigue strength exponent *b* does not change during heat treatment and may be considered as a constant value for a given steel.

The LCF tests performed for three different steels suitable for heat treatment have shown the significant influence of the process parameters (tempering temperatures) on the hardness and cyclic stress–strain and strain–life characteristics. For all steels: C45, X20Cr13, and 34CrNiMo6, qualitative relationships between tempering temperatures and the material characteristics were very similar. Lower tempering temperatures applied in the process resulted in higher values of the cyclic stress–strain branches and the elastic part of strain–life characteristics.

For each type of steel, elastic parts of the strain–life characteristics remain parallel after being subjected to different heat treatment processes, which means that the fatigue strength exponent *b* does not change. Such a material behavior may be explained considering two phenomena related to the microstructural changes in volume and with changes in the microstructure due to the Q and T processes. The former may produce additional residual stresses and the latter may change local conditions where plastic deformations occur. It is interesting to note that, in the case of X20Cr13 steel, the coefficient values relating equi-biaxial stresses to the changes of indentation force Δ*P** presented in [17] and the structural stress *σ** vs. Δ*P**, in the present work, are different. This means that in the second case, both effects occur simultaneously, resulting in increases in both the yield stresses and elastic parts of the strain–life fatigue characteristics.

The plastic parts of the strain–life characteristics diminish and rotate (Figure 11), while the material hardness increases, which means that plastic strength exponent *c* changes due to the heat treatment. Unfortunately, the structural stress parameter *σ** cannot be directly used in this case. However, the plastic part of the displacement amplitude *ε_a__p_*, which applies in the Manson–Coffin–Basquin formula, can be derived from the Ramberg–Osgood equation, considering the known elastic strain component.

Rotations of, and decreases in, the plastic components of the fatigue characteristics are more difficult to explain, considering their qualitative features. However, it can be observed that lower positions of the plastic strain amplitudes occur with accompanying higher stress amplitudes. This fact is omitted while analyzing the plastic strain amplitude vs. fatigue life. Therefore, additional experimental and analytical work was necessary in order to explain this phenomenon.

## 6. Conclusions

An analysis of experimental results yielded the conclusions that heat treatment processes have a significant influence on hardness, cyclic stress–strain and strain–life characteristics. With lower tempering temperatures, higher hardness and higher branches of the cyclic stress–strain characteristics were obtained. These results agree with some observations reported in the literature [34,35] dealing with changes in fatigue properties due to the high temperature produced during laser welding.

The characteristic parameter ∆*P**, given by Equation (5), was introduced to show the analogy between the quantitative determination of stress values by the method described in [17] and the value of the hypothetical structural stress *σ** occurring as a result of heat treatment. However, while in the first case, the stress *σ** corresponded to the existing equi-biaxial stress, in the present situation, this allowed us to shift the elastic part of the strain–life characteristic, that is, to change the parameter *σ’_f_* to the proper value resulting from the experiment.

The proposed indentation force *P** and its changes Δ*P** seem to be promising in identifying the *σ** parameter necessary for correcting values of the fatigue strength coefficients *σ_f_’* for various tempering temperatures. However, the constant indentation depth *h** = 4 μm, assumed in the present study, was based on the previous results carried out for X20Cr13 steel that have been used in comparative studies. Future work may show that the *h** value should be changed.

## Figures and Tables

**Figure 1 materials-17-02375-f001:**
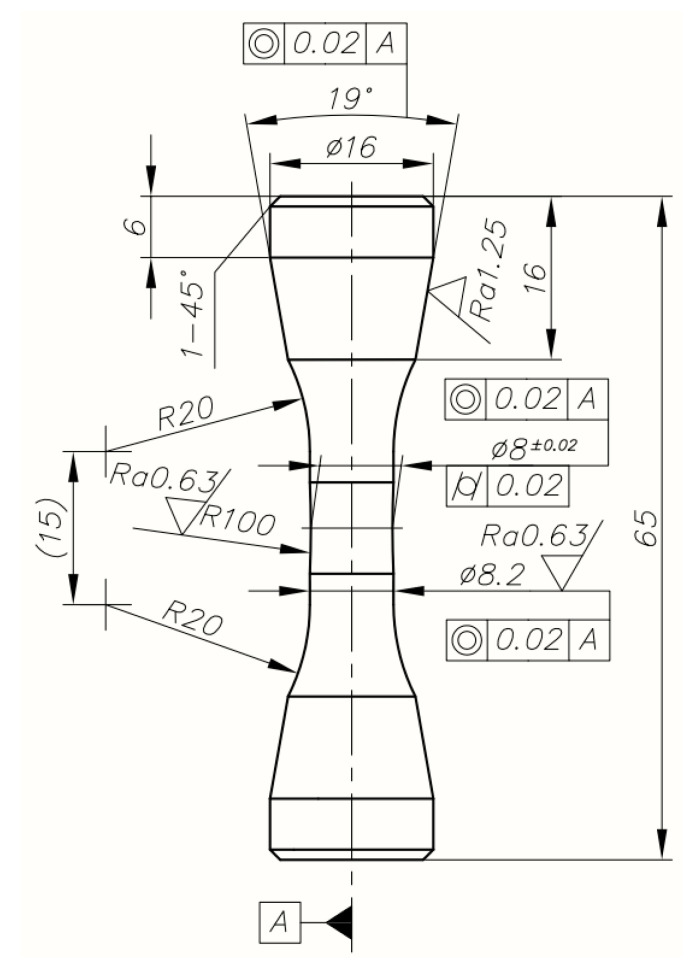
Geometrical details of specimens used in LCF tests.

**Figure 2 materials-17-02375-f002:**
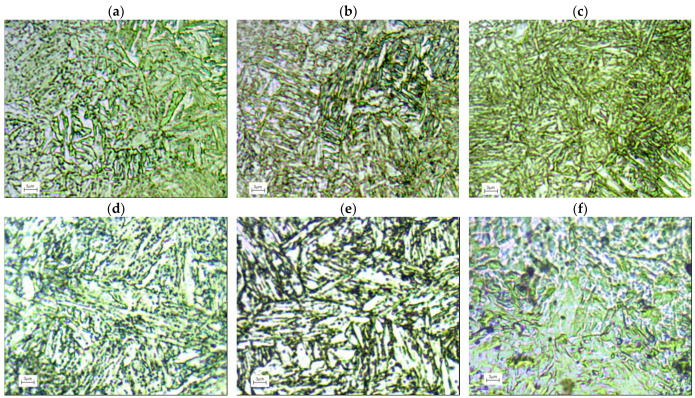
Microstructure of different steel samples after heat treatment (**a**–**c**) C45; (**d**–**f**) X20Cr13; and (**g**–**i**) 34CrNiMo6 steel used in final experiments.

**Figure 3 materials-17-02375-f003:**
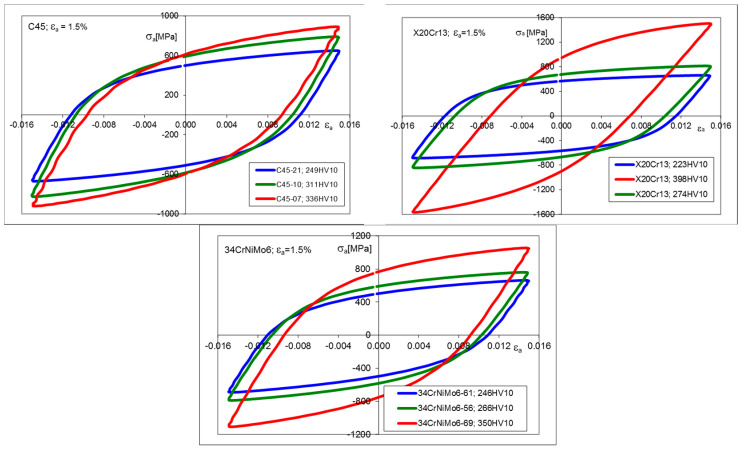
Stress–strain hysteresis loops for C45, X20Cr13, and 34CrNiMo6 steels, corresponding to 50% *N_f_*, for *ε_a_* = 1.5%.

**Figure 4 materials-17-02375-f004:**
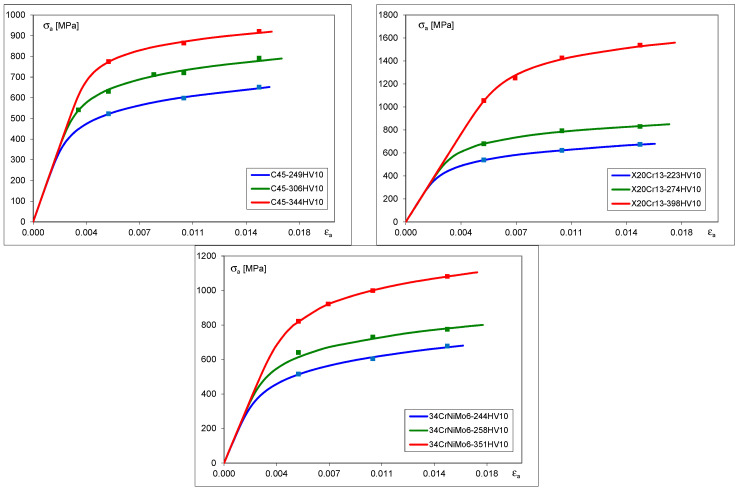
Cyclic stress–strain curves for C45, X20Cr13 and 34CrNiMo6 steels.

**Figure 5 materials-17-02375-f005:**
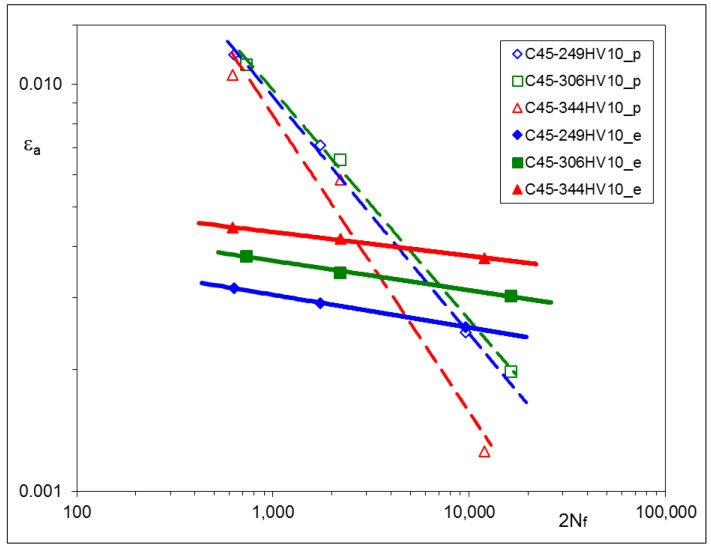
Strain–life characteristic of C45 steel subjected to various Q and T processes.

**Figure 6 materials-17-02375-f006:**
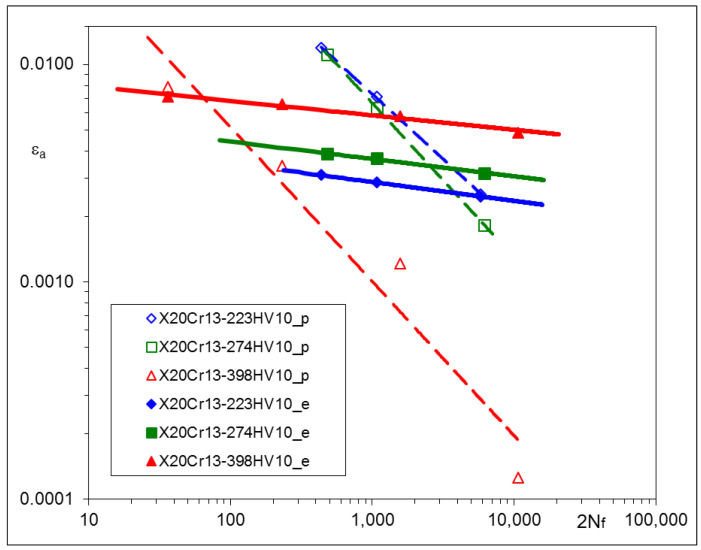
Strain–life characteristic of X20Cr13 steel subjected to various Q and T processes.

**Figure 7 materials-17-02375-f007:**
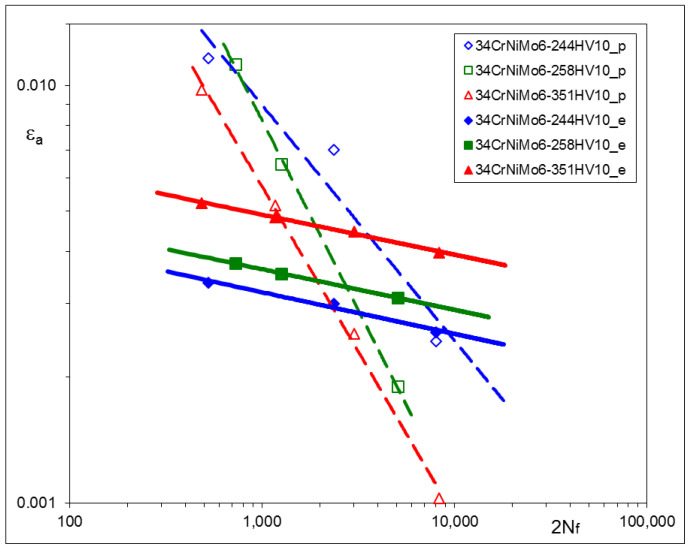
Strain–life characteristic of 34CrNiMo6 steel subjected to various Q and T processes.

**Figure 8 materials-17-02375-f008:**
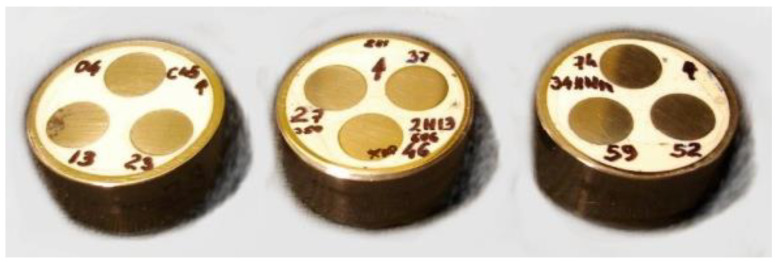
Specimens prepared for deriving characteristic values of forces *P** by means of indentation using Vickers probe.

**Figure 9 materials-17-02375-f009:**
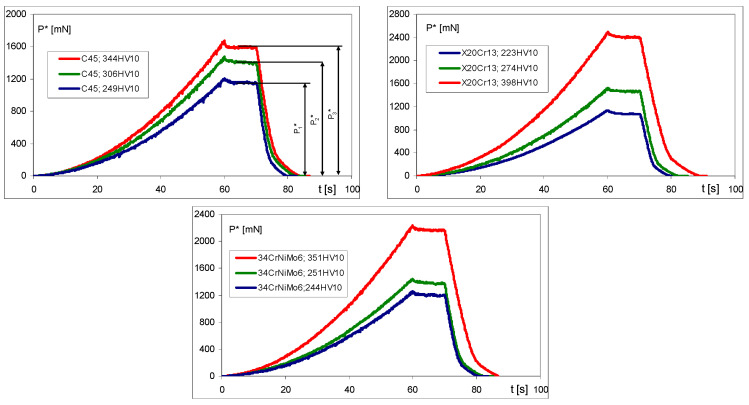
Examples of indentation graphs for C45, X20Cr13, and 34CrNiMo6 steels.

**Figure 10 materials-17-02375-f010:**
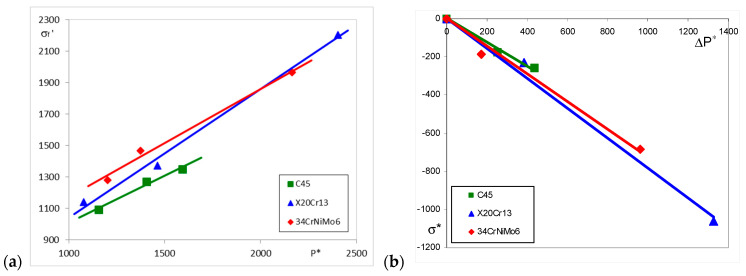
Relationship between the indentation force *P** and the fatigue strength coefficient *σ_f_′* (**a**), and between the structural stress *σ** and the increase in indentation force Δ*P** due to the heat treatment (**b**).

**Figure 11 materials-17-02375-f011:**
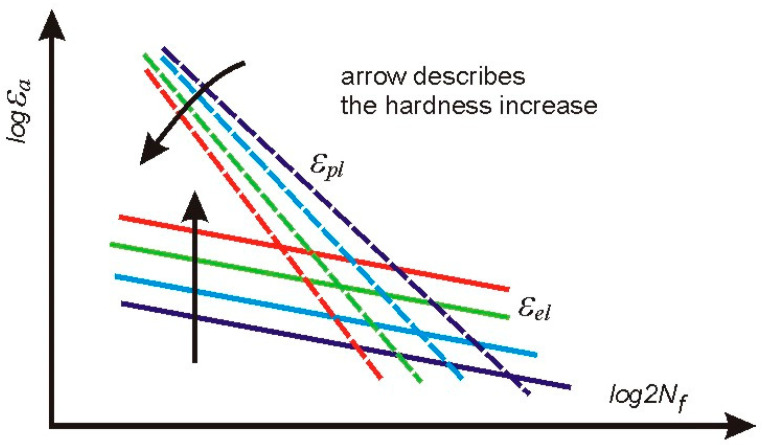
General qualitative behavior of elastic and plastic parts of the strain–life characteristics found for all steels subjected to heat treatment.

**Table 1 materials-17-02375-t001:** Mechanical properties of C45, X20Cr13, and 34CrNiMo6 steels [18].

Steel	Yield Strength R_p0.2_, (MPa)	Tensile Strength R_m_, (MPa)	Elongation A_5_, (%)	Hardness HB
C45	310–505	565–585	12–16	163–170
X20Cr13	345	655	25	195
34CrNiMo6	min. 900 (Q + T)	1100–1300 (Q + T)	min. 10 (Q + T)	<248 (+A)

**Table 2 materials-17-02375-t002:** Chemical composition of steels (wt.%).

Steel	C	Mn	Si	Ni	Cr	Cu	Mo	Al	Sn	V	Fe
C45	0.44	0.54	0.25	0.22	0.12	0.12	-	-	-	-	98.31
X20Cr13	0.19	0.45	0.44	0.39	11.61	0.10	0.04	-	-	0.06	86.72
34CrNiMo6	0.35	0.52	0.29	1.44	1.47	0.22	0.16	0.02	0.02	-	95.51

**Table 3 materials-17-02375-t003:** Heat treatment parameters and hardness obtained from preliminary Q and T tests.

Quenching (°C)	Tempering (Air) (°C)	Average Hardness	
		HRc	HV10
C45			
850 (water)	-	59 ± 1	670 ± 10
	460	38 ± 1	344 ± 18
	520	31 ± 1	306 ± 4
	620	23 ± 1	249 ± 4
X20Cr13			
1000 (oil)	-	52 ± 1	562 ± 8
	550	45 ± 1	398 ± 22
	620	30 ± 1	274 ± 18
	700	21 ± 1	223 ± 4
34CrNiMo6			
850 (oil)	-	50 ± 1	590 ± 10
	500	43 ± 1	351 ± 19
	620	30 ± 1	258 ± 19
	700	25 ± 1	244 ± 11

**Table 4 materials-17-02375-t004:** LCF test results of C45 steel.

Specimen Number	Hardness HV10	Total Strain Amplitude ε*_a_*	Number of Cycles to Failure *N_f_* (Cycles)	Plastic Strain Amplitude ε*_ap_*	Elastic Strain Amplitude ε*_ae_*	Stress Amplitude σ*_a_* _(0.5_ *_Nf_*_)_ (MPa)
C45-22	245 ± 4	0.005	4790	0.0025	0.0025	523
C45-20	253 ± 3	0.010	763	0.0071	0.0029	598
C45-21	249 ± 2	0.015	317	0.0118	0.0032	650
C45-15	298 ± 6	0.003	15,619	0.0004	0.0026	541
C45-11	305 ± 7	0.005	8089	0.0020	0.0030	630
C45-12	305 ± 6	0.008	1392	0.0046	0.0034	712
C45-14	307 ± 6	0.010	1098	0.0065	0.0035	721
C45-10	311 ± 1	0.015	363	0.0112	0.0038	792
C45-08	332 ± 4	0.005	5976	0.0013	0.0037	775
C45-03	365 ± 6	0.010	1100	0.0058	0.0042	864
C45-07	336 ± 6	0.015	310	0.0106	0.0044	921

**Table 5 materials-17-02375-t005:** LCF test results of X20Cr13 steel.

Specimen Number	Hardness [HV10]	Total Strain Amplitude ε*_a_*	Number of Cycles to Failure *N_f_* (cycles)	Plastic Strain Amplitude ε*_ap_*	Elastic Strain Amplitude ε*_ae_*	Stress Amplitude σ*_a_* _(0.5_ *_Nf_*_)_ (MPa)
X20Cr13-38	222 ± 3	0.005	2921	0.0025	0.0025	538
X20Cr13-40	228 ± 1	0.010	537	0.0071	0.0029	623
X20Cr13-41	220 ± 4	0.015	218	0.0119	0.0031	674
X20Cr13-31	253 ± 2	0.005	3084	0.0018	0.0032	679
X20Cr13-32	282 ± 2	0.010	538	0.0063	0.0037	793
X20Cr13-30	287 ± 8	0.015	242	0.0111	0.0039	830
X20Cr13-45	375 ± 5	0.005	5371	0.0001	0.0049	1055
X20Cr13-42	439 ± 6	0.007	782	0.0012	0.0058	1252
X20Cr13-43	388 ± 2	0.010	116	0.0034	0.0066	1426
X20Cr13-44	388 ± 9	0.015	18	0.0079	0.0071	1537

**Table 6 materials-17-02375-t006:** LCF test results of 34CrNiMo6 steel.

Specimen Number	Hardness HV10	Total Strain Amplitude ε*_a_*	Number of Cycles to Failure *N_f_* (Cycles)	Plastic Strain Amplitude ε*_ap_*	Elastic Strain Amplitude ε*_ae_*	Stress Amplitude σ*_a_* _(0.5_ *_Nf_*_)_ (MPa)
34CrNiMo6-60	243 ± 7	0.005	4001	0.0024	0.0026	515
34CrNiMo6-66	244 ± 4	0.010	1181	0.0070	0.0030	604
34CrNiMo6-61	246 ± 5	0.015	262	0.0116	0.0034	678
34CrNiMo6-58	249 ± 6	0.005	2537	0.0019	0.0031	640
34CrNiMo6-57	260 ± 9	0.010	631	0.0065	0.0035	729
34CrNiMo6-56	266 ± 4	0.015	364	0.0112	0.0038	775
34CrNiMo6-70	350 ± 4	0.005	4156	0.0010	0.0040	821
34CrNiMo6-68	376 ± 4	0.007	1505	0.0025	0.0045	921
34CrNiMo6-67	328 ± 2	0.010	585	0.0052	0.0048	999
34CrNiMo6-69	350 ± 9	0.015	242	0.0098	0.0052	1081

**Table 7 materials-17-02375-t007:** Strain-stress characteristic parameters obtained from LCF tests.

Material and Hardness	*K*′ (MPa)	*n*′	*E* (GPa)	*R* ^2^
C45-249HV10	1191.7	0.1377	206.7	0.996
C45-306HV10	1289.3	0.1122	207.4	0.989
C45-344HV10	1310.9	0.0791	206.6	0.991
X20Cr13-223HV10	1277.1	0.1447	217.8	0.999
X20Cr13-274HV10	1388.4	0.1128	218.0	0.992
X20Cr13-398HV10	2378.7	0.0917	216.7	0.994
34CrNiMo6-244HV10	1440.1	0.1718	200.9	0.989
34CrNiMo6-258HV10	1460.0	0.1398	206.6	0.915
34CrNiMo6-351HV10	1899.1	0.1216	206.8	0.999

**Table 8 materials-17-02375-t008:** Strain-life characteristic parameters obtained from LCF tests.

Material and Hardness	*σ_f_*′/*E*	*ε_f_*′	*b*	*c*	*E* (GPa)
C45-249HV10	0.0053	0.5236	−0.0706	−0.5829	206.7
C45-306HV10	0.0061	0.4763		−0.5641	207.4
C45-344HV10	0.0065	1.2920		−0.7291	206.6
X20Cr13-223HV10	0.0052	0.4593	−0.0818	−0.5992	217.8
X20Cr13-274HV10	0.0064	0.8916		−0.7092	218.0
X20Cr13-398HV10	0.0102	0.4547		−0.8609	216.7
34CrNiMo6-244HV10	0.0064	0.4422	−0.0976	−0.5642	200.9
34CrNiMo6-258HV10	0.0071	4.4059		−0.9093	206.6
34CrNiMo6-351HV10	0.0095	1.3324		−0.7892	206.8

**Table 9 materials-17-02375-t009:** Characteristic indentation force *P** for various materials subjected to heat treatment.

Material	Average Hardness HV10	Indentation Force *P** (mN)
C45	249 ± 2	1154 ± 9
	306 ± 6	1406 ± 5
	344 ± 5	1592 ± 6
X20Cr13	223 ± 3	1077 ± 16
	274 ± 2	1463 ± 25
	398 ± 6	2405 ± 40
34CrNiMo6	244 ± 4	1201 ± 14
	251 ± 7	1373 ± 16
	351 ± 4	2165 ± 9

**Table 10 materials-17-02375-t010:** Relationship between the structural stress parameter *σ** and range of indentation force Δ*P** resulting from the heat treatment.

Material	Structural Stress Parameter *σ** (MPa) Changes of Indentation Force Δ*P***_i_* = *P***_i_* − *P**_1_ (mN)
C45	$\sigma^{*}=-0.621\cdot \Delta P^{*}$ R^2^ = 0.980
X20Cr13	$\sigma^{*}=-0.784\cdot \Delta P^{*}$ R^2^ = 0.991
34CrNiMo6	$\sigma^{*}=-0.724\cdot \Delta P^{*}$ R^2^ = 0.983

## Data Availability

Data are contained within the article.

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
