# Peer review of "Application of Instrumented Indentation Procedure in Assessing the Low-Cycle Fatigue Properties of Selected Heat-Treated Steels"

_materials, 2024, doi:10.3390/ma17102375_

Round 1

Reviewer 1 Report

Comments and Suggestions for Authors

The paper explores the low-cycle fatigue (LCF) properties of C45, X20Cr13, and 34CrNiMo6 steels after various heat treatments with strain-controlled LCF tests at strain amplitudes of 0.5%, 1%, and 1.5%. Then, it investigates how heat treatment affects fatigue life, cyclic stress-strain behavior, and hardness, and establishes a correlation between these properties and a new structural force parameter, P*, determined using a Vickers indenter. This parameter, along with its changes ΔP*, shows potential for predicting the structural stress parameter σ* that aids in calculating fatigue strength coefficients for different tempering temperatures. However, the following comments should be addressed before this manuscript can be considered for publication,

1. The authors should enhance the introduction of this manuscript. Firstly, the introduction is currently quite succinct and lacks a comprehensive review of recent advancements in the field. Furthermore, since several existing studies have already investigated the relationship between LCF and indentation, the authors need to provide more detailed elaboration on this relationship and the specific motivations driving this study.

2. In section 2, the authors mentioned the steels used for experiments since these steels of different mechanical properties and chemical compositions are commonly used in mechanical engineering. The authors should provide the basic mechanical properties similar to the chemical composition of steels in Table 1.

3. In line 85, the authors mentioned that some preliminary tests for all materials were carried out. The authors should specifically list the types of tests performed, rather than using the vague term "some preliminary tests."

4. The tempering time used in the study is only 20 minutes, which is shorter than the typical duration for tempering steel. The authors should provide further justification for choosing such an annealing and tempering period in the manuscript. Also, the authors should provide a brief sample preparation process for the microstructure observation of different steel samples in Figure 2.

5. Since strain values used in the manuscript are quite large (e.g., 1.5%). The authors should include a complete stress-strain curve from initial loading to finial failure so that the material degradation process can be clearly observed.

6. Eq (3) indicates that the coefficient parameter b remains constant across different tempering temperatures. The authors should elaborate on the basis for this determination, specifically explaining whether it is based on experimental observations or if it is a theoretical assumption.

7. What is the purpose of ΔP*, which considers the difference between the indentation force for the material with different tempering temperatures? As different tempering temperatures can significantly change material properties, materials treated at different temperatures might behave as distinct materials. Thus, the rationale behind calculating ΔP* here is not apparent to the reviewer and requires further explanation.

8. A conclusion section should be provided for this manuscript in addition to the discussion section.

Author Response

Dear Reviewer,

Thank you very much for your thorough and substantive review.

As authors of the paper, we were aware that the proposed method of determining the low-cycle fatigue properties of heat-treated materials using the instrumented indentation procedure may require additional clarification. Based on the assumption that the potential reader should receive as complete and clear a picture as possible of the issue under discussion, we tried to meet these requirements. Consequently, on the one hand, the work came to be quite long (too long in our opinion). On the other hand, it may not contain all the explanations expected by the reader. Therefore, we know that any comments and suggestions will strike the right balance between the volume and quality of information contained in the body of the work.

Answering your comments and suggestions:

  1. The Introduction was enlarged.

The inspiration for writing the current paper, was presented in the introduction section from line 55 to line 82, of the original version.

  1. Mechanical properties of steels considered in the paper are added in Table 1.

3 i 4a. Because Question 3 and the first part of Question 4 are related, we will answer both together. The preliminary tests were aimed at determining such values of tempering temperatures that the hardness of the material between groups of samples would be between 5 and 15 HRc. The second condition to be met by 8 mm diameter specimens was uniform micro-hardness across the transversal section. However, it was not the intension of the treatment to eliminate residual stresses, so the annealing time could be shorter than generally applied. It was important to maintain the same tempering time for all specimens. Preliminary tests also included measurements of micro-hardness along the cross-sections of particular specimens. All tests have proved constant micro-hardness values along diameter of the specimen cross sections. Therefore we considered 20 minutes period acceptable.   

4b. Samples were ground with sandpaper of 600 to 4000 grit, then polished with an aqueous suspension of Al2O3 0.05mm abrasive powder until a mirror-like surface was obtained. Surfaces prepared in this way were etched with nital (a solution of nitric acid in alcohol), in the case of X20Cr13 steel - with a solution of HNO3 + HF.

  1. The changes in the shape of the stress-strain hysteresis loops of individual materials for the first, 0.5Nf and final cycles are shown in the Appendix. Placing all load cycles on the figure may make it unreadable. Therefore we placed additional figure A11 showing the whole maximum cyclic stress smax history during fatigue tests for all materials, while εa = 1.5%.
  2. The parallel location of the elastic part of the strain-life characteristic εa = f(Nf) resulting in the constant value of the exponent b of steels subjected to different heat treatments, was first found by analyzing the experimental results of one of the steels, namely C45 steel. Two other steels were selected to investigate if a similar phenomenon would also occur in different steels. The positive result obtained for all selected steels was a base to formulate theoretical description of the problem together with a reliable measurement procedure.

  1. The characteristic parameter ∆P*, given by Equation (5), was introduced to show the analogy between the quantitative determination of stress values by the method described in the paper [17] and the value of the hypothetical structural stress σ* occurring as a result of heat treatment. However, while in the first case the stress determined by the method described in [17] corresponded to the existing equi-biaxial stress, in the present situation the parameter σ* allows us to shift the elastic part of the strain-life characteristic, that is, to change the parameter σ'f to the proper value resulting from the experiment.

The meaning of the introduction of the parameter ∆P* is that the σ* value is determined relative to a known “reference” heat treatment conditions for which σ* = 0, and the force P* > 0 is known. Therefore, in any other case (for a different tempering temperature than that of the reference material), the σ* value corresponds to the ∆P*, which represents the difference between the actual and the reference force.

  1. Conclusions are added.

Best regards,

Authors

Reviewer 2 Report

Comments and Suggestions for Authors

Congratulations on your work. You have combined well the experimental results with the analytical computational relations.

Did you have any problems when measuring the strain with the extensometer during fatigue stress? What happened when failure was reached and near failure?

Perhaps it would be good if you provided some photos of the broken specimens.

There are a few small corrections to be made which are marked in yellow and specified in the comments in the attached file.

In the attached file there are a few observations that please consider. From figure 4 (which does not exist) you need to renumber.

Author Response

Dear Reviewer,

Thank you very much for your in-depth and substantive review.

As authors of the paper, we were aware that the proposed method of determining the low-cycle fatigue properties of heat-treated materials using the instrumented indentation procedure may require additional clarification. Based on the assumption that the potential reader should receive as complete and clear a picture as possible of the issue under discussion, we tried to meet these requirements. Consequently, on the one hand, the work came to be quite large (too large in our opinion). On the other hand, it may not contain all the explanations expected by the reader. Therefore, we know that any comments and suggestions will strike the right balance between the volume and quality of information contained in the body of the work.

Answering your comments and suggestions:

An extensometer with a base of 12.5 mm was used to measure specimen displacement, and it was placed on the cylindrical surface of the specimen. For cyclic loading occurring in fatigue tests, two types of problems can occur. First, the sharp edges of the extensometer can imprint small notches on the lateral surface of the specimen, which, through the “notch effect”, may become the initial site of crack formation. Second, slippage of the extensometer blade can occur in samples with significant hardness. To prevent this, small amounts of resin were applied to the extensometer attachment points on the specimen. Samples in which cracks occurred near the blades were rejected.  

Some photos of the broken specimens are added in Appendix.

All suggested corrections have been considered in the text.

Picture numbers were changed accordingly.

Best regards,

Authors

Round 2

Reviewer 1 Report

Comments and Suggestions for Authors

By carefully checking the revised manuscript and corresponding response, the comments proposed by the reviewer are addressed appropriately. 

From the reviewer's perspective, the content quality is improved and the manuscript is recommended for publication in its present form.